# LEARNING REWARDS AND SKILLS TO FOLLOW COMMANDS WITH A DATA EFFICIENT VISUAL-AUDIO REPRESENTATION

## ABSTRACT

Based on the recent advancements in representation learning, we propose a novel framework for command-following robots with raw sensor inputs. Previous RL-based methods are either difficult to continuously improve after the deployment or require a large number of new labels during the fine-tuning. Motivated by (self-)supervised contrastive learning literature, we propose a novel representation, named VAR++, that generates an intrinsic reward function for command-following robot tasks by associating images with sound commands. After the robot is deployed in a new domain, the representation can be updated intuitively and data-efficiently by non-experts, and the robot is able to fulfill sound commands without any hand-crafted reward functions. We demonstrate our approach on various sound types and robotic tasks, including navigation and manipulation with raw sensor inputs. In the simulated experiments, we show that our system can continually self-improve in previously unseen scenarios given fewer new labeled data, yet achieves better performance, compared with previous methods.

## 1 INTRODUCTION

When humans are told to turn on a TV, they can associate what they hear with what they see even in unfamiliar environments. For robots to follow commands and fulfill similar tasks, they must ground task-oriented language to vision and motor skills. Command following robots is such an important application that paves the way for non-experts to intuitively communicate and collaborate with robots in daily lives.

The need for command following robots has spurred a wealth of research. Learning-based language grounding agents were proposed to perform tasks according to visual observations and text/speech instructions Anderson et al. (2018); Chang et al. (2020); Chaplot et al. (2018); Hermann et al. (2017); Shridhar et al. (2020); Yu et al. (2018). However, these approaches fail to completely solve a common problem in learning-based methods: performance degradation in a novel target domain Akkaya et al. (2019); James et al. (2019); Tobin et al. (2017). One solution to address the domain shift problem is domain randomization Tobin et al. (2017). However, it has been shown that domain randomization alone is not sufficient since the randomized simulation may not accurately reflect the target domain that the robot is later deployed in Du et al. (2021); Smith et al. (2022). Alternatively, fine-tuning policies in the target domain can further reduce the reality gap but is often cost prohibitive: professionals usually train the robots with hand-crafted, task-specific *reward functions* Haarnoja et al. (2019); Smith et al. (2022) and large amounts of *labels*, neither of which can be afforded by non-expert users after deployment.

Without enough domain expertise or abundant labeled data, how can we allow users to adapt such robots to novel domains with minimal supervision? Prior works have partially answered this question by proposing a visual-audio representation (VAR) trained with triplet loss, which associates audio commands and goal images with the same intent Chang et al. (2021). However, due to true negative pairs used in the triplet loss, the number of labels required to fine-tune the VAR is still not satisfactory, thus hindering an efficient deployment in the target domain.

In this paper, we propose a novel framework that builds on (self-)supervised contrastive learning to realize more *effective* training and more *efficient* fine-tuning for rewards and skills learning. As

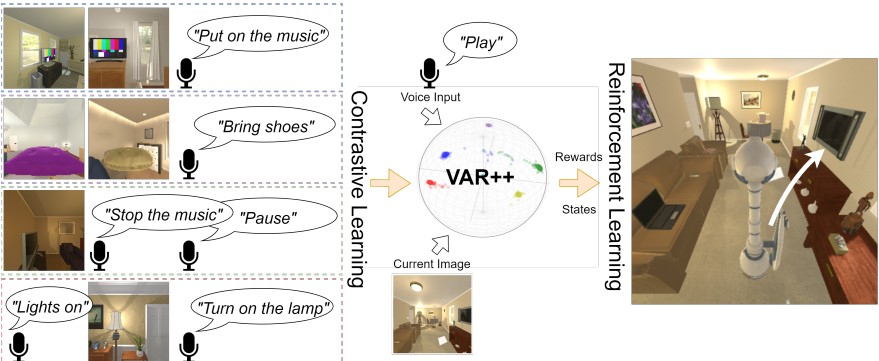

Figure 1: **Illustration of our pipeline.** Contrastive learning is used to group images and audio commands of the same intent. The resulting representation VAR++ supports the downstream RL training by encoding the high-dimensional voice and image signals, and providing reward signals and states to the agent.

shown in Fig. 1, we first learn a joint representation of visual and audio signals (VAR++) whose clusters have better intra-cluster cohesion and inter-cluster separation compared to VAR Chang et al. (2021). In the second stage, we use VAR++ to compute intrinsic reward functions to learn various robot skills with RL without any reward engineering. When the robot is deployed in a new domain such as a different room, the fine-tuning stage is data efficient in terms of the label usage and is natural to non-experts in terms of the human-robot interaction. For example, a user can teach a robot or VAR++ by saying that "this is an apple" when the robot sees an apple. Then, RL policies are self-improved with the updated VAR++. No hand-designed reward or negative pairs are needed as in the previous works.

We apply this learning approach to different robotic tasks in diverse settings as illustrated in Fig. 1 and Fig. 3. Given a sound command, the robot must identify the commander's goal (intent), draw the correspondence between the raw visual and audio inputs, and develop a policy to finish the task. The tasks are challenging because no maps, depth images, human demonstrations, or prior knowledge are available, and the observation mainly comes from a monocular uncalibrated RGB camera.

Our main contributions are as follows: (1) We propose a novel representation of visual-audio observations for command following robots, named VAR++. To our best knowledge, it is the first work demonstrating that (self-)supervised contrastive loss improves *robot control* performance from triplet loss. (2) We propose a data efficient fine-tuning method for command following robots which requires significantly fewer labels than baselines. Moreover, our fine-tuning method demonstrates that (self-)supervised contrastive loss has the potential to enhance user experiences, especially for non-experts. (3) We release our simulation environments and model implementations. The simulations are the first open-sourced AI environments which use real speech recordings for robotic command following tasks. The code will be released after the review at `www.github.com`

## 2 RELATED WORKS

**End-to-end language understanding.** End-to-end spoken language understanding (SLU) systems extract speaker's intent directly from raw speech signals without translating the speech to text Kim et al. (2021); Lugosch et al. (2019); Serdyuk et al. (2018). Such an end-to-end system is able to fully exploit subtle information, such as speaker emotion, that is lost during speech to text transcription Kim et al. (2021); Lugosch et al. (2019). However, end-to-end SLU systems are mainly developed for virtual digital assistants and not for robotic applications.

**Language grounding agents.** Conventional language grounding agents consist of independent modules for transcription, language grounding, and planning Magassouba et al. (2019); Paul et al. (2018); Stramandinoli et al. (2016). These modular pipelines suffer from intermediate errors and do not generalize beyond their programmed domains Hermann et al. (2017); Tada et al. (2020); Vanzo et al. (2016). To address these problems, end-to-end language grounding agents are used to perform tasks according to text-based natural language instructions and visual observations Anderson et al. (2018); Chaplot et al. (2018); Hermann et al. (2017); Shridhar et al. (2020); Yu et al. (2018). Our

work has two novelties from these works. First, while the above works consider text-based input, we focus on commanding agents through raw audio, which leads to more natural human-robot communication without additional modules. Second, training the agents requires either expert demonstrations, step-by-step instructions Anderson et al. (2018); Shridhar et al. (2020), or a carefully designed extrinsic reward function Chaplot et al. (2018); Hermann et al. (2017); Yu et al. (2018). Although some methods generalize to new command sentences and/or new scenes to some extent, they overlook the continual fine-tuning after deployment Chang et al. (2020); Yu et al. (2018). In contrast, our method requires none of the above and thus requires significantly fewer efforts to fine-tune in a novel domain, where both perception and dynamics are different from the training scenes.

Another line of works trains the robot to fulfill commands directly from raw audio inputs with RL Chang et al. (2020; 2021). However, the method in Chang et al. (2020) requires hand-tuned reward functions and a prohibitive number of one-hot labels, which is still hard to fine-tune. Chang et al. (2021) partially addresses the problem by learning a visual-audio representation (VAR) with triplet loss to generate an intrinsic reward function for RL. However, the quality of the VAR in Chang et al. (2021) is suboptimal based on our quantitative evaluation. In contrast, the better representation in our work leads to more ideal reward functions and better robot performance.

**Representation learning for robotics.** Representation learning has shown great potential in learning useful embeddings for downstream robotic tasks. Deep autoencoders have been used to compress high-dimensional observations such as images into low-dimensional latent space. The resulting latent vectors are then used as states or intrinsic rewards for RL Lange et al. (2012); Nair et al. (2018); Wang et al. (2020). At test time, however, the methods in Nair et al. (2018); Wang et al. (2020) require users to provide goal images for task execution, while our method takes voice commands, which is a more natural and convenient way of human-robot communication. Additionally, reconstructions of the input images often make the autoencoders computationally expensive. Another line of works uses contrastive loss to learn representations for downstream tasks such as grasping and water pouring Jang et al. (2018); Nguyen et al. (2020); Sermanet et al. (2018). Contrastive loss avoids the reconstruction operation in autoencoders. While all of these works focus mainly on the visual or the text modality, we address the interplay between sight and sound.

## 3 METHODOLOGY

In this section, we describe the two-stage training pipeline and fine-tuning procedure. In training, we assume the availability of sufficiently large labeled datasets, simulators, and labels. However, in fine-tuning, speech transcriptions, one-hot labels, and reward functions, are not available.

### 3.1 VISUAL-AUDIO REPRESENTATION LEARNING

In the first stage, we collect visual-audio pairs from the environment. Then, we learn a joint representation of images and audios, named VAR++, that associates an image with its corresponding sound command.

**Data collection.** Suppose there are $M$ possible intents or tasks within an environment. We collect visual-audio pairs defined as $(\mathbf{I}, \mathbf{S}, y)$ from the environment, where $\mathbf{I} \in \mathbb{R}^{n \times n}$ is the current RGB image from the robot's camera, $\mathbf{S} \in \mathbb{R}^{l \times m}$ is the Mel Frequency Cepstral Coefficients (MFCC) Davis & Mermelstein (1980) of the sound command, and $y \in \{0, 1, ..., M\}$ is the intent ID. We call $\mathbf{I}$ and $\mathbf{S}$ two *views* of an intent $y$. A visual-audio pair contains an image and a sound command of the same intent. For example, when an iTHOR agent sees a lit lamp, it hears the sound "Switch on the lamp" from the environment. In contrast, when the agent sees no object or is far away from all objects so that it sees many objects at once, it receives only an image and hears no sound. The image is paired with $\mathbf{S} = \mathbf{0}_{l \times m}$ and $y = M$. We define this situation as an *empty intent*.

**Training VAR++.** Our goal is to encode both visual and auditory signals into a joint latent space, where the embeddings from the same intents are pulled closer together than embeddings from different intents. For example, the embedding of an image with a TV turned on needs to be close to the embedding of a sound command "Turn on the TV" but far away from other irrelevant commands such as "Turn off the light." We adopt the idea from (self-)supervised contrastive learning for visual representations and formulate the problem as metric learning. As shown in Fig. 2a, the VAR++ is a double-branch network with two main components. The first component is the en-

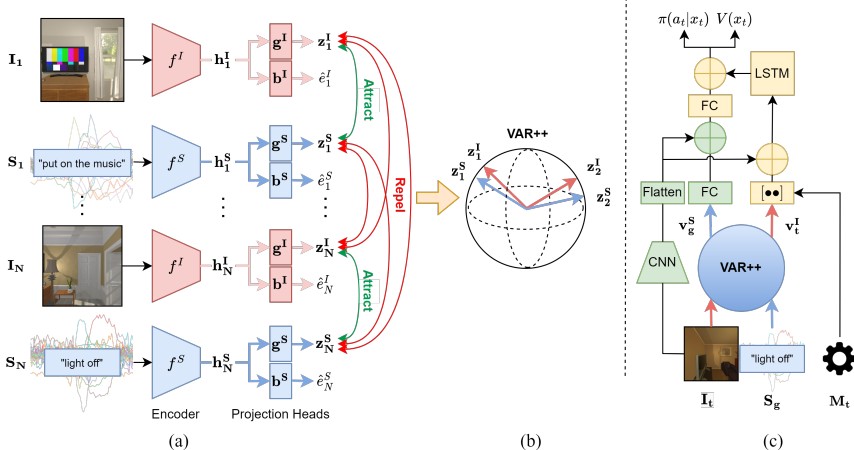

(a)  (b)  (c)

Figure 2: **Network architectures**. (a) The VAR++ is a double-branch network optimized with (self-)supervised contrastive loss. (b) The latent space of the VAR++ is a unit hypersphere such that the images and audios of the same intent are closer than those of different intent in the space. (c) The policy network for RL training. The portion in blue is VAR++ which is frozen during the RL training. We use $\bigoplus$ to denote element-wise addition, $FC$ to denote fully connected layers, and $[\bullet\bullet]$ to denote concatenation.

coders $f^I : \mathbb{R}^{n \times n} \to \mathbb{R}^{d_I}$ and $f^S : \mathbb{R}^{l \times m} \to \mathbb{R}^{d_S}$ which map an input image $\mathbf{I}$ and a sound signal $\mathbf{S}$ to representation vectors $\mathbf{h}^I$ and $\mathbf{h}^S$, respectively. In practice, any deep models for image and sound processing can be used for $f^I$ and $f^S$. The second component is the projection heads $g^I : \mathbb{R}^{d_I} \to \mathbb{R}^d$, $g^S : \mathbb{R}^{d_S} \to \mathbb{R}^d$, $b^I : \mathbb{R}^{d_I} \to \mathbb{R}$, and $b^S : \mathbb{R}^{d_S} \to \mathbb{R}$ that map the representations $\mathbf{h}^I$ and $\mathbf{h}^S$ to the space where losses are applied. We denote the vector embeddings $g^I(\mathbf{h}^I)$ and $g^S(\mathbf{h}^S)$ as $\mathbf{z}^I$ and $\mathbf{z}^S$, respectively. We enforce the norm of $\mathbf{z}^I$ and $\mathbf{z}^S$ to be 1 by applying an $L2$-normalization, such that the embeddings live on a unit hypersphere as shown in Fig. 2b.

We use supervised contrastive (SupCon) loss as the objective, which encourages the distance between $\mathbf{z}^I$ and $\mathbf{z}^S$ of the same intent to be closer than those of a different intent Khosla et al. (2020). Suppose there are $N$ visual-audio pairs in a batch. Let $k \in K := \{1, ..., 2N\}$ be the index of an image or a sound signal within that batch and $P(k) := \{p \in K \setminus \{k\} : y_p = y_k\}$ be the set of indices of all images and sounds of the same intent except for index $k$. Then, the SupCon loss is

$$\mathcal{L}_{\text{SupCon}} = -\sum_{k \in K} \frac{1}{|P(k)|} \sum_{p \in P(k)} \log \frac{\exp\left(\mathbf{z}_k \cdot \mathbf{z}_p / \tau\right)}{\sum_{j \in K \setminus \{k\}} \exp\left(\mathbf{z}_k \cdot \mathbf{z}_j / \tau\right)}, \quad (1)$$

where $|\cdot|$ is the cardinality, $\mathbf{z}_{(\cdot)}$ can be either $\mathbf{z}^I$ or $\mathbf{z}^S$, and $\tau \in \mathbb{R}^+$ is a scalar temperature parameter. The previous VAR uses visual-audio triplets of the form $(\mathbf{I}, \mathbf{S}^+, \mathbf{S}^-)$ for the training, where $\mathbf{I}$ and $\mathbf{S}^+$ are an image and sound with the same intent, and $\mathbf{S}^-$ is the sound with a different intent. The loss only pulls together the embeddings of $\mathbf{I}$ and $\mathbf{S}^+$ and pushes away the embeddings of $\mathbf{I}$ and $\mathbf{S}^-$ in a triplet. This setting is less efficient because each anchor only has one positive and one negative Chang et al. (2021). In contrast, the use of SupCon loss allows the attraction and repulsion among all images and sound within a batch, which improves the performance of the representation as we will show in Sec. 4.3. We additionally introduce a binary classification loss for both the image and sound to distinguish between empty and non-empty intent. Let $\mathcal{L}_{\text{BCE}}$ denote the binary cross entropy loss and $e$ denote the label of intent, which is 0 for empty intent and 1 for non-empty intent. The batch loss for training the VAR++ is:

$$\mathcal{L}_{\text{VAR++}} = \alpha_1 \mathcal{L}_{\text{SupCon}} + \alpha_2 \frac{1}{N} \sum_{j=1}^{N} \mathcal{L}_{\text{BCE}}(b^I(\mathbf{h}_j^I), e_j) + \mathcal{L}_{\text{BCE}}(b^S(\mathbf{h}_j^S), e_j), \quad (2)$$

where $\alpha_1$ and $\alpha_2$ are the weights of losses. Depending on if the intent is predicted empty or not, the output $\mathbf{v}^I$ and $\mathbf{v}^S$ of VAR++ can be determined for image and sound by:

$$\begin{aligned}
\mathbf{v}^I &= \mathbb{1}_{\{\hat{e}^I \geq 0.5\}} \mathbf{z}^I, \quad \hat{e}^I := b^I(\mathbf{h}^I), \\
\mathbf{v}^S &= \mathbb{1}_{\{\hat{e}^S \geq 0.5\}} \mathbf{z}^S, \quad \hat{e}^S := b^S(\mathbf{h}^S),
\end{aligned} \quad (3)$$

where $\mathbb{1}$ is an indicator function. We call the latent space of the output as *joint space*. The purpose of the binary classification is to set the image and sound embeddings of the empty intent to the center

of the joint space. This centralization removes the biases caused by the location of the empty intent in the joint space, leading to better intrinsic reward generated by the VAR++.

While SupCon loss and other self-supervised visual representation learning frameworks are originally only applied to image modality Chen et al. (2020); Khosla et al. (2020), we extend the framework to a multi-modality setting and create a new representation for command following robots.

## 3.2 RL WITH VISUAL-AUDIO REPRESENTATION

The second stage of our pipeline is to train an RL agent using an intrinsic reward function generated by a trained VAR++. We model a robot command following task as a Markov Decision Process (MDP), defined by the tuple $\langle \mathcal{X}, \mathcal{A}, P, R, \gamma \rangle$. At each time step $t$, the agent receives an image $\mathbf{I}_t$ from its RGB camera, and robot states $\mathbf{M}_t$ such as end-effector location or previous action. At $t = 0$, an additional one-time sound command $\mathbf{S}_g$ containing an intent is given to the robot. We freeze the trainable weights of VAR++ in this stage and define the MDP state $x_t \in \mathcal{X}$ as $x_t = [\mathbf{I}_t, \mathbf{v}_t^I, \mathbf{v}_g^S, \mathbf{M}_t]$, where $\mathbf{v}_t^I$ and $\mathbf{v}_g^S$ are the output of the VAR++ for $\mathbf{I}_t$ and $\mathbf{S}_g$, respectively. The VAR++ encodes the information in the input image and the intent in $\mathbf{S}_g$. Then, based on its policy $\pi(a_t|x_t)$, the agent takes an action $a_t \in \mathcal{A}$. In return, the agent receives a reward $r_t \in R$ and transitions to the next state $x_{t+1}$ according to an unknown state transition $P(\cdot|x_t, a_t)$. The process continues until $t$ exceeds the maximum episode length $T$, and the next episode starts.

**Intrinsic rewards.** Since $\mathbf{v}^I$ and $\mathbf{v}^S$ of the same intent are pulled together within the VAR++ by the contrastive loss, intrinsic rewards can be derived as the similarity between $\mathbf{v}^I$ and $\mathbf{v}^S$. Eq. 4 and 5 present two possible task-agnostic and robot-agnostic reward functions:

$$r_t^i = \mathbf{v}_t^I \cdot \mathbf{v}_g^S \tag{4}$$

$$r_t^{ic} = \mathbf{v}_t^I \cdot \mathbf{v}_g^S + \mathbf{v}_t^S \cdot \mathbf{v}_g^S \tag{5}$$

where $\mathbf{v}_t^S$ is the embedding of the current sound signal $\mathbf{S}_t$, which can be triggered in the same way as $\mathbf{S}$ as described in Section 3.1. Intuitively, the agent using $r_t^i$ receives high reward when the scene it sees matches the command it hears. The agent trained using the reward $r_t^{ic}$ additionally needs to match the current sound it hears with the sound command to receive high rewards. Compared to $r_t^{ic}$, the reward function $r_t^i$ does not depend on any real-time supervision signal such as current sound $\mathbf{v}_t^S$ from the environment, allowing the agent to perform *self-supervised* RL training with VAR++. Although RL agents trained with Eq. 4 can already achieve decent performance, providing the current sound $\mathbf{S}_t$ can further improve the performance Chang et al. (2021). Since $\mathbf{S}_t$ can be difficult to obtain especially in real environments, $\mathbf{S}_t$ is not part of the state $x_t$ and thus the robot policy does not require $\mathbf{S}_t$ at test time.

**Policy network architecture.** We show our policy network architecture used in our experiments in Fig. 2c. Given the state $x_t$, the network outputs the value $V(x_t)$ and the policy $\pi(a_t|x_t)$. Instead of reusing the CNN in $f^I$, we add another CNN to extract the features relevant for achieving the goal. For example, the iTHOR agent needs to encode information about obstacles for collision avoidance. We use an LSTM to encode the embeddings of $\mathbf{I}_t$ and $\mathbf{M}_t$ for long-term decision making. Proximal Policy Optimization (PPO) was used for policy and value function learning Schulman et al. (2017).

## 3.3 FINE-TUNING

After the robot is deployed in a new domain such as the real world, its performance often degrades due to domain shift from both perception and dynamics Du et al. (2021). Our fine-tuning procedure allows non-experts to *continually* improve the VAR++ to reduce perception gaps and improve robot skills to reduce dynamics gaps. Since performing accurate state and reward measurements, data labeling, and instrumentation requires domain expertise and is time-consuming, we assume tuned reward functions, one-hot labels, and accurate speech transcriptions are not available from non-experts. Fortunately, our method requires none of these. To fine-tune VAR++, since we no longer have the underlying labels $y$ for images and sounds, we replace the SupCon loss in Eq. (2) with the following self-supervised contrastive loss (SSC) Chen et al. (2020):

$$\mathcal{L}_{\text{SSC}} = -\sum_{k \in K} \log \frac{\exp\left(\mathbf{z}_k \cdot \mathbf{z}_{p(k)}/\tau\right)}{\sum_{j \in K \setminus \{k\}} \exp\left(\mathbf{z}_k \cdot \mathbf{z}_j/\tau\right)}, \tag{6}$$

where $p(k)$ is the index of the data paired with the data of index $k$ with the same intent. To fine-tune the robot, non-experts collect visual-audio pairs of the form $(\mathbf{I}, \mathbf{S})$ based on their common knowledge using their own voices. The robot can then self-improve its policy network with the intrinsic reward function by randomly sampling a collected sound command as the goal. See Appendix A for a detailed fine-tuning algorithm.

To fine-tune VAR in Chang et al. (2021), non-experts have to provide a sound command with different intent $\mathbf{S}^-$ for each image $\mathbf{I}$ to use triplet loss. In contrast, VAR++ eliminates this requirement by utilizing the SSC, leading to a more intuitive data collection experience for non-experts and better performance with fewer labeled data.

## 4 SIMULATION EXPERIMENTS

In this section, we first describe the environments (Fig. 1) and various sound datasets for the experiments. Then, we compare the performance and data efficiency of our pipeline with several baselines and ablation models.

### 4.1 ROBOTIC ENVIRONMENTS

We evaluate the performance of all the methods on three different robotic platforms: Turtle-Bot, Kuka, and iTHOR. In all environments, after hearing a sound command, the robots must learn exploration skills and approach the corresponding objects. All the robots are equipped with a monocular uncalibrated RGB camera for robot perception. See Appendix C for detailed descriptions and visualizations.

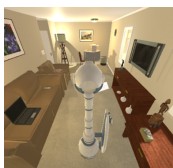 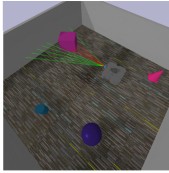 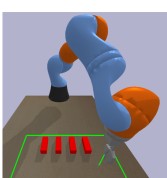

iTHOR environment    TurtleBot environment    Kuka environment

Figure 3: Simulation environments for the experiments.

### 4.2 SOUND DATA

We use several types of sounds from state-of-the-art datasets in training and testing. Specifically, we use speech signals collected for training the end-to-end SLU from Fluent Speech Commands (FSC) Lugosch et al. (2019) and short speech commands from Google Speech Commands (GSC) Warden (2018). We also collect single-tone signals from NSynth Engel et al. (2017) and urban & environmental sounds from UrbanSound8K (US8K) Salamon et al. (2014). The *Wordset dataset* was created from the "0," "1," "2," "3" in GSC. We also used a *Mix dataset* to show that the VAR++ can map multiple types of sounds to a single object or idea, by mixing speech data with environmental sound. We mix "house" with "jackhammer" and "dog" with "bark". Examples of commands in the iTHOR environment include "turn on the lights" and "pause". The iTHOR environment uses the commands from FSC, while the Kuka and the TurtleBot environments uses the commands from the other sound datasets. See Appendix B for more sound examples and intent we choose for the iTHOR environment.

### 4.3 EVALUATION OF THE REPRESENTATIONS

**Evaluation metrics.** The representations are evaluated by a linear layer (LL) and nearest neighbor (NN). For LL, we follow the widely used linear evaluation protocol, where a linear classifier is trained using a cross-entropy loss on top of the frozen encoders, which are $f^I$ and $f^S$ in our case Chen et al. (2020); Kolesnikov et al. (2019); Zhang et al. (2016). For NN, we first find the medoids of each intent $C_0, ..., C_M$ in the joint space using the training data. The predicted label of a test image or sound is $\arg\max_i \mathbf{v} \cdot C_i$, where $\mathbf{v}$ is embedding of the test image or sound in the joint space. NN measures the quality of the intrinsic reward produced by the representations.

**Baselines.** We compare the performance of our VAR++ with VAR which uses triplet loss for both training and fine-tuning Chang et al. (2021). For each intent, we collect the same number of visual-audio pairs $(\mathbf{I}, \mathbf{S}, y)$ for VAR++ training and visual-audio triplets of the form $(\mathbf{I}, \mathbf{S}^+, \mathbf{S}^-)$ for VAR training, where $\mathbf{I}$ and $\mathbf{S}^+$ are an image and sound with the same intent, $y$ is the intent ID, and $\mathbf{S}^-$ is the sound with a different intent. We kept the network architecture of both methods the same.

**Quantitative results.** From Table 1, we observe that both methods achieve high NN accuracy while VAR++ marginally outperforms VAR, suggesting that both methods are able to produce accurate and reliable rewards for the downstream RL tasks. As for LL, VAR++ is much better than VAR since even a linear classifier can achieve much higher accuracy with VAR++.

**Qualitative results.** We visualize the VARs by projecting images and sounds to the joint space, as shown in Fig. 4. We see that the embeddings of the same concept form a cluster and all clusters are separated from each other. Compared

Table 1: Percentage accuracy of VARs with nearest neighbor (NN) and linear layer (LL).

| Env | Method | NN | LL | | |
|---|---|---|---|---|---|
| | | | Img | Snd | Avg |
| Kuka | VAR | 97.8 | 77.4 | 95.4 | 86.4 |
| | VAR++ | **98.5** | **82.2** | **99.8** | **91.0** |
| TurtleBot | VAR | 96.7 | 66.1 | 84.9 | 75.5 |
| | VAR++ | **99.1** | **76.5** | **96.8** | **88.1** |
| iTHOR | VAR | 96.6 | 51.4 | 91.2 | 73.8 |
| | VAR++ | **96.9** | **78.4** | **94.7** | **86.6** |

to VAR, the clusters in VAR++ have better intra-cluster cohesion and inter-cluster separation, suggesting that the two distinct concepts are better distinguished and the same concepts are better related. During fine-tuning, although VAR++ does not have $S^-$ as an explicit indication of negatives like VAR does in the input, VAR++ can still maintain relatively clear inter-cluster separation and provide reliable rewards for the self-improvement of RL agents.

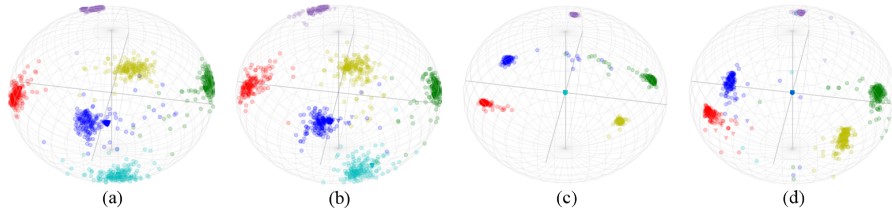

| (a) | (b) | (c) | (d) |

Figure 4: **Visualizations of the VARs in the iTHOR environments with FSC.** The colors indicate the ground truth intent ID of embeddings of sound (marked by triangles) and image (marked by circles). (a) VAR after the training. (b) VAR after the fine-tuning. (c) VAR++ after the training. (d) VAR++ after the fine-tuning.

## 4.4 EVALUATION OF THE RL POLICY

**Evaluation metrics.** We evaluate the model with two metrics: (1) success rate (SR) and (2) the number of labels used for training (LU). We define SR as the percentage of successful test episodes. We test the learned policy for 50 episodes for each intent. For the iTHOR environment, an agent succeeds if it fulfills the command. For the TurtleBot and Kuka environments, a successful episode happens when the agent stays close to the target mentioned in the command for a certain time period. We compare the label usage of a model because a command following robot deployed in the real world should require as few annotations as possible from non-experts for fine-tuning.

**Baselines and ablations.** We compare the RL performance of our method against the following baselines and ablation models. The first baseline, denoted as "E2E," is a representative end-to-end deep RL policy for command following robots Chang et al. (2020). E2E uses hand-tuned task-specific reward functions and requires ground-truth class labels for image and sound classification.

The second baseline, denoted as "VAR," trains an RL agent based on the output of the VAR Chang et al. (2021). VAR utilizes triplet loss for the training and fine-tuning. Both our method and VAR use Eq 5 for the downstream RL tasks. We mark a model with "Centered (C)" or "Not centered (NC)"to indicate if the image and sound embeddings of the empty intent are set to the center of the hypersphere in the joint space. The original VAR method does not centralize the empty intent.

The third baseline, denoted as "ASR+NLU+RL (ANR)," is a common modular pipeline. We first use an off-the-shelf automatic speech recognition (ASR) named Mozilla DeepSpeech Hannun et al. (2014) to transcribe the speech to text. We then train a learning-based natural language understanding (NLU) module to handle the noisy output from the ASR. For example, "Play the music" is sometimes transcribed as "by the music." Finally, a vision-based RL agent operates with the predicted intent from the NLU. Note that unlike this baseline, our method does not rely on any transcriptions or expertise to be fine-tuned. This baseline does not work with non-speech datasets such as NSynth. See Appendix D for more facts.

**Definition of labels.** In this paper, labels include all forms of annotation and measurement that are used to train a model. For example, one-hot labels for image and sound classification and the distance measurement between the robot and the goal are both labels. One visual-audio pair $(\mathbf{I}, \mathbf{S}, y)$ for training or $(\mathbf{I}, \mathbf{S})$ for fine-tuning used in VAR++ requires 1 label to indicate $y$ or the same intent. A visual-audio triplet used in VAR, $(\mathbf{I}, \mathbf{S}^+, \mathbf{S}^-)$, requires 2 labels to indicate the positive and the negative.

Table 2: Test success rate results in Kuka and TurtleBot environments with different sounds.

| Env | Dataset | SR↑ | | | |
|---|---|---|---|---|---|
| | | ANR | E2E | VAR | Ours(C) |
| Kuka | Wordset | 85.5 | 95.5 | 97.0 | **99.0** |
| | NSynth | - | 92.5 | **98.0** | **98.0** |
| | Mix | - | 94.0 | 95.5 | **97.0** |
| TurtleBot | Wordset | 82.0 | 92.0 | 94.0 | **95.0** |
| | NSynth | - | 95.0 | 96.0 | **97.5** |
| | Mix | - | 87.0 | 91.0 | **92.5** |

Every E2E training step requires 3 labels, including the target object state checking (e.g. check if the light is switched on), distance measuring to calculate the extrinsic reward, and a one-hot label for auxiliary losses.

**Control policies with unheard sounds.** In this experiment, we test the performance of different models with sound commands never heard by the agent during training (e.g. new speakers). All the models were trained with the same number of RL steps and sufficient labels. For iTHOR environment, we trained the agents for 9 million (M) RL steps and tested them within the seen floor plans (Floor Plan 201 - 220). For Turtlebot and Kuka environments, the total RL steps is 3M. No fine-tuning is performed yet.

Table 2 and Table 3 show that the application of our method is not limited to a specific robot, robotic task, or types of sound signal. In all environments, compared to the baselines, our method achieves the highest SR. In iTHOR environment, our

Table 3: Train label-usage and test success rate results in iTHOR 201-220 with FSC dataset.

| Models | LU ($\times 10^6$)↓ | SR↑ |
|---|---|---|
| ANR | 27.00 | 66.0 |
| E2E | 27.06 | 68.0 |
| VAR(NC) | 9.12 | 65.6 |
| VAR(C) | 9.12 | 69.0 |
| Ours(NC) | **9.06** | 65.8 |
| Ours(C) | **9.06** | **72.4** |

method achieves the highest SR and the lowest LU. Although no limit was imposed on LU in this experiment, ASR+NLU+RL and E2E require much more labels during the training than VAR and our method. The results also suggest that the intrinsic rewards produced by the representations are sufficient for the RL training, since VAR and our method both demonstrate satisfying performance without receiving any extrinsic rewards.

From Table 2 and 3, the SR for ASR+NLU+RL baseline is lower than most of the other methods. The main reason is that the system suffers from intermediate and cascading errors among different modules, which coincides with the findings in Chang et al. (2021); Tada et al. (2020). The last four rows of Table 3 indicate the improvement by centralizing the empty intent for both VAR and our method. This result justifies the necessity of the binary classification loss in Eq. 2. See Appendix E for examples of task execution of the agent and Appendix F for time efficiency measurements.

**Fine-tuning in novel iTHOR floor plans.** This experiment aims to show the potential of each method to be improved in a new domain. We consider the scenario where a trained household robot is purchased to serve in a new room with a unfamiliar set of furniture and arrangement. Each method

Table 4: Average success rates over unseen iTHOR Floor Plan 226 - 230 after fine-tuning with additional label-usage.

| LU | 0 | | | | 2400 | | | |
|---|---|---|---|---|---|---|---|---|
| Models | ANR | E2E | VAR | Ours | ANR | E2E | VAR | Ours |
| Avg.↑ | 18.8 | 18.4 | 19.6 | 20.8 | 24.0 | 23.6 | 69.2 | **86.0** |

is given the same number of new labels, and a data efficient method should achieve the highest success rate. We first test the performance of trained models with unheard sound commands in 5 unseen iTHOR floor plans without any fine-tuning. This process uses 0 new labels. The first three columns of Table 4 show the necessity of fine-tuning: the performance of all methods drops due to the consequence of domain shift, which is a common problem for learning systems Tobin et al. (2017). We then use 2400 new labels for each unseen floor plan to fine-tune each method for that floor plan. For our method, each intent has 400 new labels on average because there are 6 intents for our iTHOR environment. We followed Sec. 3.3 to fine-tune the VAR and VAR++ and used Eq. 4 to

self-improve RL policies without current sounds. For E2E, we collect one-hot labels and use simulator queries during the fine-tuning. The fine-tuning is terminated after it reaches the label limit. See Appendix E.4 for comparison of task execution before and after the fine-tuning.

From Table 4, we find that the ANR and end-to-end method can only be improved by $5.2\%$ using 2400 labels, suggesting the inefficiency of fine-tuning E2E after deployment. The label quotas are depleted rapidly due to the inefficient use of labels for policy network fine-tuning, which leads to less RL experience. VAR and our method improve itself by $49.6\%$ and $65.2\%$, respectively, using the same amount of labels after 1M of self-supervised RL training steps.

The richer RL experience was due to the higher data efficiency of our method because the labels were used to update VAR++, and there was no label consumption during the self-supervised RL exploration. Compared to VAR, our method achieves better performance because VAR++ does not need negative pairs for fine-tuning. This property allows the VAR++ to achieve almost the same RL performance as VAR using only half as many labels, since Chang et al. (2021) reports that the SR for VAR with 5000 new labels is $84.7\%$.

Table 5: Model performance and the number of visual-audio pairs collected for the fine-tuning

| Floor Plan | RL steps | Number of Pairs | | | |
|---|---|---|---|---|---|
| | | 0 | 600 | 1200 | 2400 |
| 226 | 0M | 26.8 | - | - | - |
| | 0.08M | 41.6 | 70.8 | 72.0 | 84.0 |
| | 0.4M | 69.2 | 87.6 | 92.0 | 92.0 |
| | 1M | 77.6 | 88.4 | 93.2 | 97.2 |
| 229 | 0M | 34.4 | - | - | - |
| | 0.08M | 51.2 | 54.0 | 54.0 | 64.0 |
| | 0.4M | 57.6 | 70.4 | 77.6 | 95.2 |
| | 1M | 62.0 | 71.2 | 93.2 | 95.6 |

We further show the relation between the number of RL steps and the number of newly collected pairs in two randomly selected unseen iTHOR Floor Plans (226 and 229). In Table 5, we see that our method is still effective when the number of new pairs and the self-supervised RL steps are much fewer than 2400 and 1M - even when no new pairs are collected. More visual-audio pairs and more RL steps allow the agent to improve faster and reach higher success rates.

**Fine-tuning in new Kuka environment.** This experiment shows that our method can handle dynamics gaps and adapt to unseen objects. We first train the agent in the original Kuka environment with four identical blocks. At test time, we change the link mass, the joint friction, and parameters of the robot's PID controller. In addition, as shown in Fig. 7 in Appendix C.2, we also replace three of the blocks to a capsule, a teddy bear, and a rabbit. Without fine-tuning, our method achieves 69.5% SR. This result suggests that the VAR++ successfully encodes the most essential spatial information and can generalize to unseen objects with different shapes. We then fine-tune the agent following Sec. 3.3 with 1800 visual-audio pairs and 0.5M RL steps. The final SR raises to 96.5%, which demonstrates the adaptability of our method to novel objects and changes in dynamics.

## 5 FUTURE WORK AND DISCUSSION

In conclusion, we propose a novel visual-audio representation named VAR++ for command following robots based on the recent advancement in (self-)supervised contrastive learning. VAR++ requires much fewer labels from non-experts during fine-tuning but produces higher-quality rewards for downstream RL agents. Our results suggest that visual-language association and skill development are highly correlated and thus need to be designed together. Furthermore, we are the first to demonstrate that (self-)supervised contrastive loss has the potential in enhancing human-robot interaction (HRI) experiences. Such a natural human-robot interaction can promote human perception and adoption of robotic systems and marks one step towards practical social robot applications. However, our work encompasses the following limitations, which opens up directions for future work. (1) Empty intents may result in a sparse intrinsic reward function, which poses challenges in solving long horizon tasks. To solve this, our reward function can be combined with other intrinsic rewards Burda et al. (2019). (2) We only apply our method to vision-based command following robots in this paper. It is a promising direction to extend the method to other modalities and provide reward function for other goal-based multi-modal robot tasks.

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

## A    ALGORITHM FOR FINE-TUNING AN AGENT

This section shows the detailed algorithm for fine-tuning the VAR++ and an RL agent.

---

**Algorithm 1** Fine-tuning

---

1: Inputs: A trained VAR++ $\mathbf{V}$, and a trained policy $\pi_\theta$
2: Collect a small set of visual-audio pairs $\mathcal{D} = \{(\mathbf{I}_i, \mathbf{S}_i)\}_{i=1}^U$
3: **for** a sampled minibatch $\{(\mathbf{I}_i, \mathbf{S}_i)\}_{i=1}^N$ from $\mathcal{D}$ **do**          $\triangleright$ Fine-tune VAR++
4:      Calculate empty intent label $e_i$ by checking if $\mathbf{S}_i = \mathbf{0}_{l \times m}$
5:      Calculate image and sound embeddings: $\mathbf{h}^I, \mathbf{z}^I, \mathbf{h}^S, \mathbf{z}^S \leftarrow \mathbf{V}(\mathbf{I}_i, \mathbf{S}_i)$
6:      Calculate $\mathcal{L}_{\text{SSC}}$ by Eq. 6
7:      Calculate loss by $\mathcal{L}_{\text{finetune}} = \alpha_1 \mathcal{L}_{\text{SSC}} + \alpha_2 \frac{1}{N} \sum_{j=1}^N \mathcal{L}_{\text{BCE}}(b^I(\mathbf{h}_j^I), e_j) + \mathcal{L}_{\text{BCE}}(b^S(\mathbf{h}_j^S), e_j)$
8:      Update $\mathbf{V}$ to minimize $\mathcal{L}_{\text{finetune}}$
9: **for** $k = 0, 1, 2, ...$ **do**          $\triangleright$ Self-supervised RL fine-tuning
10:      Sample a sound command $\mathbf{S}_g$ from $\mathcal{D}$ as goal
11:      **for** $t = 0, 1, ..., T$ **do**
12:          Receive RGB image $\mathbf{I}_t$ and robot state $\mathbf{M}_t$
13:          Calculate image and sound embeddings: $\mathbf{v}_t^I, \mathbf{v}_g^S \leftarrow \mathbf{V}(\mathbf{I}_t, \mathbf{S}_g)$ by Eq. 3
14:          Calculate reward $r_t = \mathbf{v}_t^I \cdot \mathbf{v}_g^S$
15:          **if** $\mathbf{S}_t$ **then**
16:              Calculate embeddings: $\mathbf{v}_t^S \leftarrow \mathbf{V}(\mathbf{S}_t)$
17:              $r_t = r_t + \mathbf{v}_t^S \cdot \mathbf{v}_g^S$
18:          Store $\{r_t, \mathbf{I}_t, \mathbf{M}_t, \mathbf{v}_t^I, \mathbf{v}_g^S\}$ in a memory buffer $\mathcal{D}_{RL}$
19:      Update $\pi_\theta$ with data from $\mathcal{D}_{RL}$ using PPO
20:      Clear $\mathcal{D}_{RL}$
21: **return** $\mathbf{V}, \pi_\theta$

---

## B    SOUND DATA

Table 6: Sound signals used in the experiments.

| Dataset | Sound | Examples |
|---|---|---|
| FSC | activate light
deactivate light
activate music
deactivate music
bring shoes | "Turn on the lights," "Lamp on"
"Switch off the lamp," "Lights off"
"Put on the music," "Play"
"Pause music," "Stop"
"Get me my shoes," "Bring shoes" |
| GSC | "0," "1," "2," "3"
names of 4 objects | "zero," "one," "two," "three"
"house" "tree," "bird," "dog" |
| NSynth | $C_4, D_4, E_4, F_4$ | Various instruments, tempo, and volume |
| US8K | bark, jackhammer | Sound recorded in the wild |

## C    ROBOTIC ENVIRONMENT DESCRIPTIONS

The Turtlebot and Kuka environments are developed in PyBullet Coumans & Bai (2016–2019) and mainly posed challenges in fine motor control with moderate difficulty in perception. In contrast, the iTHOR environment is developed in AI2-THOR Kolve et al. (2017) and is challenging in perception with discretized and simplified control.

### C.1    TURTLEBOT

Four objects – a cube, sphere, cone, and cylinder – are placed in a $4m^2$ space. Each object has an associated intent. The goal of the robot is to navigate to the object corresponding to the given sound command based on RGB images. The robot and the four objects are placed randomly in the arena at the beginning of an episode. The robot's action is the change of desired transitional velocity and the change of desired orientation. The robot needs to develop exploration skills to discover the goal object in the shortest period.

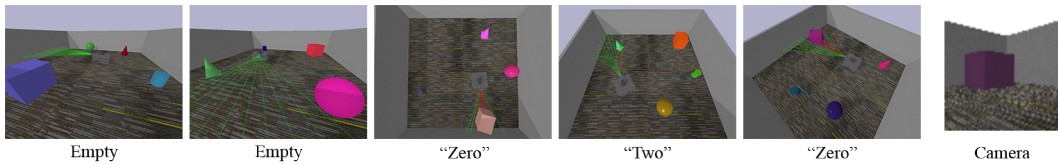

Figure 5: Visualization of the TurtleBot environment with paired images and voices from the Wordset. In this case, "zero" means cube, "one" means sphere, "two" means cone, and "three" means cylinder. The red and green rays are just for illustration purposes. The right most figure shows the camera view.

### C.2    KUKA

Four identical blocks, each associated with a sound command, are placed in a line at a random location on the table. The robot needs to move its gripper above the block corresponding to a given command based on RGB images. The camera is placed at a fixed location on the side of the table such that it can capture the gripper and blocks from a distorted perspective. The relative positions of the gripper tip and the blocks are initialized randomly at the beginning of an episode. The robot needs to develop spatial reasoning skills to approach the target block using the relative positional information observed from the camera.

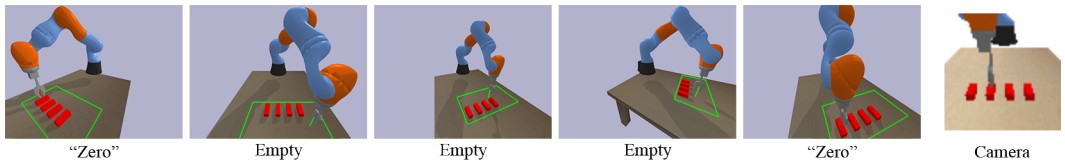

Figure 6: Visualization of the Kuka environment with paired images and voices from the Wordset. In this case, "zero" means the leftmost block, "one" means the second block from the left, and so on. The red and green rays are just for illustration purposes. The right most figure shows the camera view.

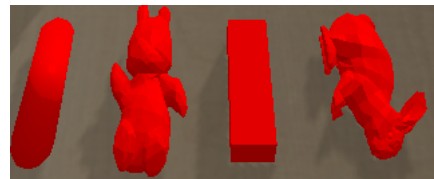

Figure 7: Visualization of the new objects for the Kuka fine-tuning experiment.

### C.3    ITHOR

Our iTHOR environment uses real full-sentence speech commands to simulate a real-world application of household robots. The environment has 30 different floor plans of living rooms, each with

their own set of decorations, furnitures, and arrangements. The robot is given goal tasks such as switching the floor lamp or television on or off. The robot must navigate through the environment and interact with the intended object given RGB images and a noisy local discrete occupancy grid as robot states. The complexity of the environment requires the agent to associate complicated speech commands with high-fidelity visual observations, without a floor plan map. The floor plans can be visualized and interact with in `https://ai2thor.allenai.org/demo/`.

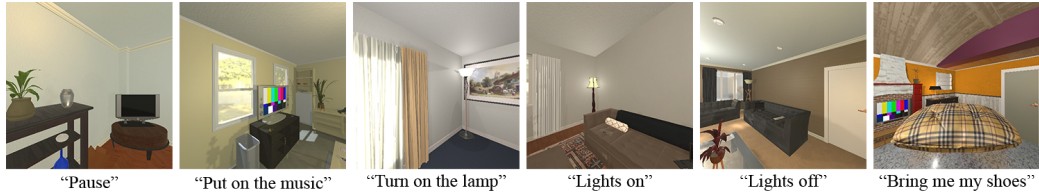

"Pause"   "Put on the music"  "Turn on the lamp"   "Lights on"   "Lights off"  "Bring me my shoes"

Figure 8: Visualization of the iTHOR environment with paired images and voices from the FSC.

## D   ABOUT ASR+NLU+RL (ANR) PIPELINE

- Accuracy of intent prediction of ASR+NLU.
  FSC dataset: 86.0%; Wordset: 87.0%.

- SR of the RL agent when given the ground-truth intent
  iTHOR environment: 79.2%; Kuka environment: 98%; TurtleBot environment: 95%.

## E   VISUALIZATION OF TASK EXECUTION

### E.1   TURTLEBOT

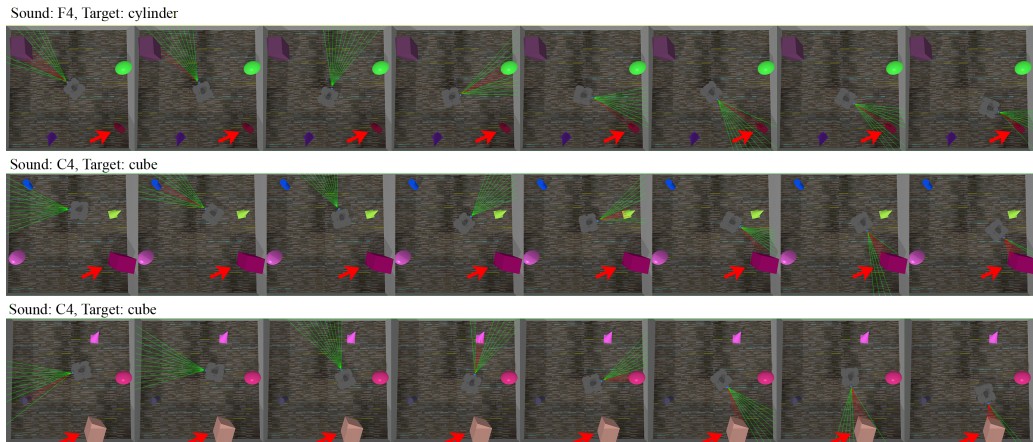

Figure 9: Visualization of the task execution in the TurtleBot environment after training without fine-tuning. The sounds come from NSynth dataset. TurtleBot searches and approaches its target successfully in all episodes.

## E.2 KUKA

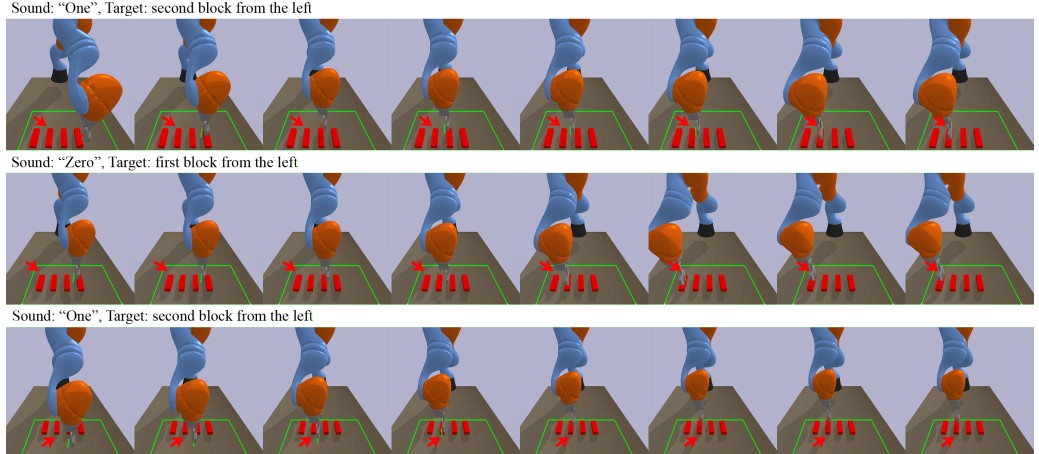

Figure 10: Visualization of the task execution in the Kuka environment after training without fine-tuning. The sounds come from Wordset dataset. Kuka moves its gripper to the target block successfully in all episodes.

## E.3 ITHOR

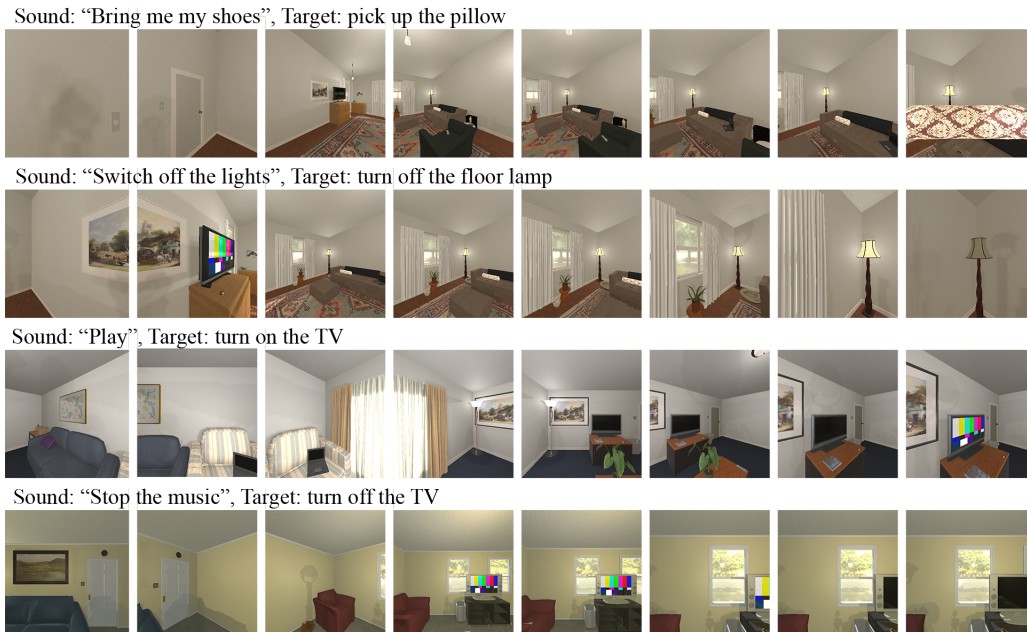

Figure 11: Visualization of the task execution in the iTHOR environment after training without fine-tuning. The sounds come from FSC dataset. iTHOR agent finishes household tasks successfully in all episodes.

### E.4 ITHOR FINE-TUNING

Sound: "Stop the music", Target: turn off the TV

Sound: "Fetch my shoes", Target: Pick up the pillow

Sound: "Turn on the lamp", Target: Turn on the floor lamp

Sound: "Turn off the lamp", Target: Turn off the floor lamp

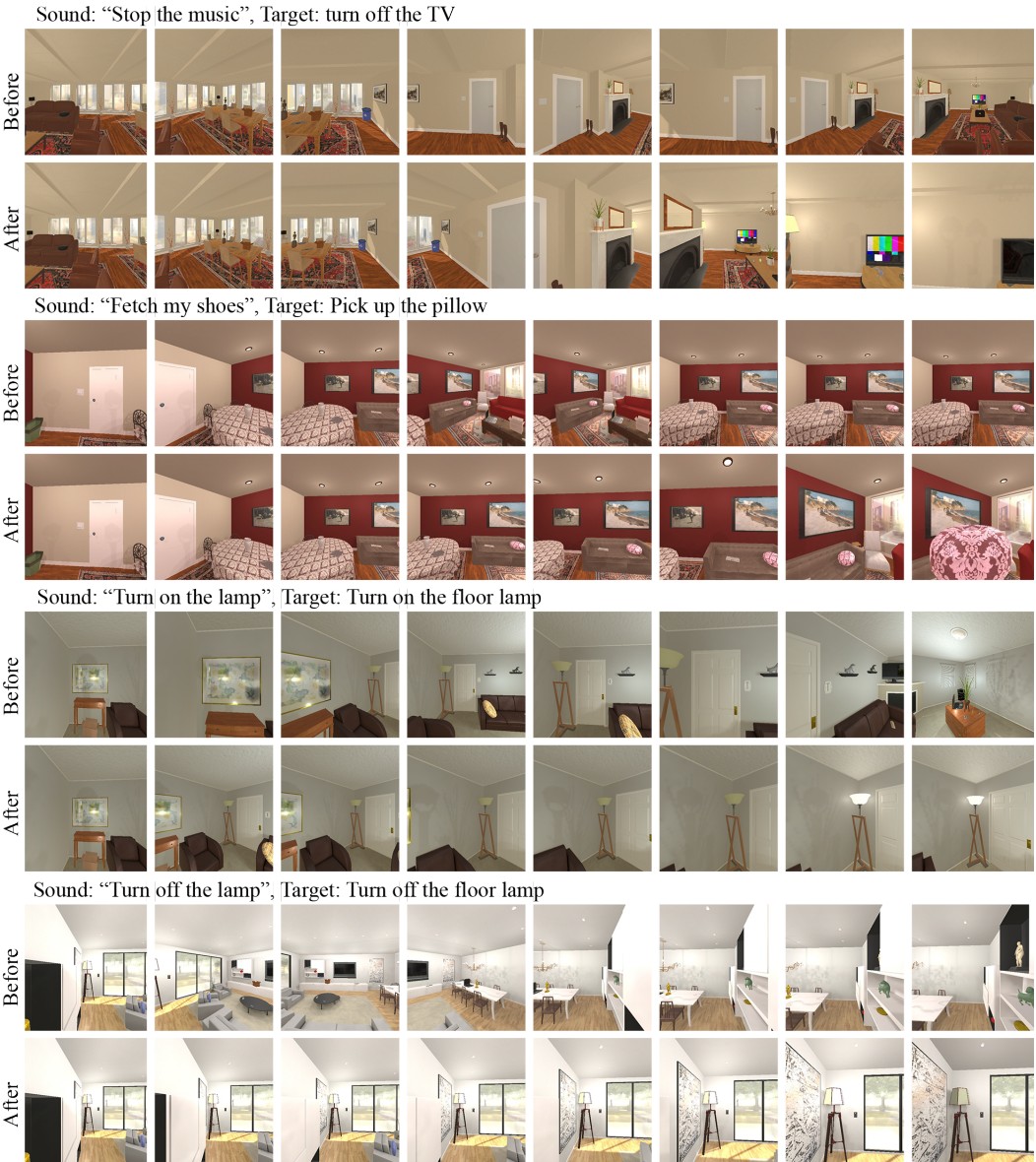

Figure 12: Visualization of the task execution in the iTHOR environment before and after the fine-tuning in unseen floor plans and the sound commands given by new speakers. The sounds come from FSC dataset.

## F   TIME EFFICIENCY

In this section, we evaluate the time efficiency of all the methods. All the models are running on a single Nvidia GTX 1080 Ti GPU and a Intel(R) Core(TM) i7-8700 CPU @ 3.20GHz. We report the average time in second (s) for the model to take one action in the iTHOR environment with the FSC dataset. The average is calculated from 12500 samples.

- ANR: 0.041s
- E2E: 0.018s
- VAR: 0.024s
- VAR++: 0.022s

