# OpenReview forum: "Learning Rewards and Skills to Follow Commands with a Data Efficient Visual-Audio Representation"
_ICLR.cc/2023/Conference — Submitted to ICLR 2023_

### Official Review · Reviewer_hTKz · 2022-10-22

**Confidence:** 4
**Correctness:** 4
**Technical Novelty And Significance:** 2
**Empirical Novelty And Significance:** 2
**Recommendation:** 5

**Clarity, Quality, Novelty And Reproducibility:**

The paper is clearly written and detailed. The appendix could be enhanced by providing all hyperparameter settings or a link to the open-source codebase.

**Strength And Weaknesses:**

Strenghs:
- very relevant problem space: this is an important area of research,
- the model is well-motivated, and described in sufficient detail to be generally useful,
- the quantitative results are compelling.

Weaknesses:
- The whole paper can be summarized as: 'we took VAR, replaced the triplet loss with a contrastive loss, and things worked better'. While a solid result to be publishing, the novelty is limited: the general idea that contrastive losses generally improve over triplets is well documented in the literature.
- The model is well engineered and reasonable for the setting that's explored, but the paper doesn't provide any unique, novel insight beyond 'here is an architecture that works well for this problem'. This means there is little one will learn from reading this paper beyond this exact setting.


**Summary Of The Paper:**

This paper proposes an architecture for audio visual representations that can be used to embed multi-model representations useful for downstream robot learning tasks.

**Summary Of The Review:**

Nice contribution to a very specific problem, which is clear and correct, but likely too incremental to meet the bar for a venue like ICLR.

---

> ### Author Response · Authors · 2022-11-17
> **Response to Reviewer hTKz**
>
> We appreciate your constructive comments. We sincerely hope that you will take into account the response we have made below and re-evaluate the merit of this paper. The modifications we made is marked by blue text in the paper.
>
> **Q1: While a solid result to be publishing, the novelty is limited: the general idea that contrastive losses generally improve over triplets is well documented in the literature.**
>
> We update our contributions in our introduction and impacts in our conclusion. We also discuss about the novelty and contribution in the "Response to all reviewers" on this webpage.
>
>
> **Q2: The model is well engineered and reasonable for the setting that's explored, but the paper doesn't provide any unique, novel insight beyond 'here is an architecture that works well for this problem'. This means there is little one will learn from reading this paper beyond this exact setting.**
>
> Interpreting visually-grounded commands has been acknowledged as an important and long-held goal of robotics and artificial intelligence [3]-[7]. We respectfully argue that this work contributes to the large community of command following robots/language grounding agents. By taking raw audios as input rather than texts in previous works, the proposed method generalizes command following robots to broader application settings. In detail, the proposed method enables robots to comprehend non-lexical vocalizations in addition to languages. For example, when a door bell rings, a VAR++ equipped robot has the ability to navigate to and open the door, while a previous text-based command following robot may fail the task as no corresponding text can be generated from the non-lexical sound. Such a potential to react to different types of sounds is a key in realizing highly intelligent agents.
>
> More broadly, the architecture of VAR++ is not restricted to command following robots and can be extended to other goal-based multi-modal robot tasks. From a high level, VAR++ works by leveraging a perception branch, which perceives the environment, and a goal inference branch, which extracts the goal that the robot is required to achieve. The input modalities of the two branches can be heterogeneous, and thus the system is multi-modal. The command following robot studied in this paper is a special case of the above setting where the input to the perception branch is RGB images and that to the goal inference branch is raw audios. A similar framework can still be used, for example, when the robot uses LiDAR point clouds as the perception signal and texts as the goal indicator. As a result, researchers can use VAR++ as a template and baseline to build a wide range of applications. Extending VAR++ to broader applications is a direction of our future work and we believe that a close analysis and design of the representation presented in this paper is a preliminary step to achieve our goal.
>
> **Q3: The appendix could be enhanced by providing all hyperparameter settings or a link to the open-source codebase.**
>
> We enhance the appendix with visualizations and facts. We open-source our codebase for better reproducibility. We attached the code as a supplementary material as well.
>
>
> References
>
> [1] Prannay Khosla, et al., “Supervised Contrastive Learning,” NeurIPS, 2020.
>
> [2] Beliz Gunel, et al., “Supervised Contrastive Learning for Pre-trained Language Model Fine-tuning,” ICLR, 2021.
>
> [3] Devendra Singh Chaplot, et al., “Gated-Attention Architectures for Task-Oriented Language Grounding”, AAAI, 2018.
>
> [4] Haonan Yu, et al., “Interactive Grounded Language Acquisition and Generalization in a 2D World”, ICLR 2018.
>
> [5] Aly Magassouba, et al., “Understanding Natural Language Instructions for Fetching Daily Objects Using GAN-Based Multimodal Target–Source Classification”, RA-L, 2019.
>
> [6] Mohit Shridhar, et al., “ALFRED: A Benchmark for Interpreting Grounded Instructions for Everyday Tasks”, CVPR, 2020.
>
> [7] Peter Anderson, et al., “Vision-and-Language Navigation: Interpreting visually-grounded
> navigation instructions in real environments”, CVPR 2018.

---

> > ### Comment · Reviewer_hTKz · 2022-11-29
> > **Thank you for the replies**
> >
> > And in particular for providing the code for reproducibility.
> > Taking all the reviewer comments into consideration, my current evaluation of the paper still stands.

---

### Official Review · Reviewer_Fs4r · 2022-10-23

**Confidence:** 3
**Correctness:** 3
**Technical Novelty And Significance:** 2
**Empirical Novelty And Significance:** 2
**Recommendation:** 6

**Clarity, Quality, Novelty And Reproducibility:**

The paper is generally written clearly, including the formulation of the problem and of the proposed solution. The novelty is, in my understanding, although interesting, marginal. The method introduced mainly adds on top of an existing method. The ideas proposed to improve the existing method are interesting, although I cannot evaluate their novelty with respect to other existing works.
The results are not reproducible, as there are very little details of the implemented models and of the simulated environments used.

**Strength And Weaknesses:**

The paper is well structured and written in a clear way. The problem of interest is introduced and motivated in a comprehensive way, mentioning relevant literature in the field. The "list of contributions" reads more as the list of steps describing the approach and its advantages, rather than highlighting the novelties of this particular piece of work. The methodology is presented in a clear way and the diagrams are clarifying and well presented as well. The experimental setup is also described fairly well, however there is very little visualisation (only Fig. 4) of the simulated environment and setup. There is also no visualisation of the data and of the results obtained. The tables do report quantitative numbers, however they seem sometimes incomplete: for example why is ASR+NLU+RL not included in Table 2 and 4? Also, what is the impact of the transcription step (which is not part of the ASR+NLU+RL per se) on the performance obtained from applying ASR+NLU+RL? What are the results if you used (correct) text directly?
The limitations of the proposed work are not discussed. Can you describe in which cases the proposed method would fail or perform just as good as other methods? Is there an evaluation on time efficiency? Is the proposed method also more efficient with respect to other alternatives?


**Summary Of The Paper:**

This paper presents VAR++, that is representation that generates intrinsic rewards by associating images and sound commands. This model is used to perform command-following robot tasks. The method builds on top of VAR (Visual-audio representation), is evaluated in 3 different tasks in simulation, and is compared against other 2 baselines in addition to the basic version.

**Summary Of The Review:**

The proposed method is interesting and the paper well written. The novelty and contribution of the paper are incremental. The experimental evaluation does not include enough details to be reproducible, and the quantitative evaluation is missing some results (eg relative to one of the baselines considered) and some discussion about the limitations of the proposed method.

---

> ### Author Response · Authors · 2022-11-17
> **Response for Reviewer Fs4r**
>
> We thank the reviewer for the comments and suggestions. We add the missing results and visualisations to the paper. The modifications we made is marked by blue text in the paper.
>
> **Q1: The "list of contributions" reads more as the list of steps describing the approach and its advantages, rather than highlighting the novelties of this particular piece of work.**
>
> We have updated our list of contributions to highlight the novelties of this work at the end of Sec. I.
>
> **Q2: There is very little visualisation (only Fig. 4) of the simulated environment and setup. There is also no visualisation of the data and of the results obtained.**
>
> We take your suggestion into account and add richer visualisations to our paper. Specifically, Appendix C contains the descriptions and visualisations of the simulated environment and setup. Appendix E contains the visualisations of task execution of the agent in each of our environments, as well as comparisons of task execution before and after the fine-tuning.
>
> **Q3: The tables do report quantitative numbers, however they seem sometimes incomplete: for example why is ASR+NLU+RL not included in Table 2 and 4? Also, what is the impact of the transcription step (which is not part of the ASR+NLU+RL per se) on the performance obtained from applying ASR+NLU+RL? What are the results if you used (correct) text directly?**
>
> To solve the incompleteness, we add the evaluation of this baseline into both Table 2 and Table 4. Since the NSynth dataset and the Mix dataset contain non-speech contents, we mainly evaluate this baseline with Wordset and the FSC dataset.
> We also add more facts about this baseline to Appendix D. For example, the accuracy of the transcription step is 86% on the FSC dataset. If given ground-truth intent (correct text, assuming perfect NLU), this oracle RL agent has an SR of 79.2% in the iTHOR environment. However, one of the drawbacks of this baseline is that it requires ground-truth text transcriptions and expertise in speech recognition and language modeling to be fine-tuned, which we assume unavailable during the fine-tuning. The negative impact from the transcription step has motivated the end-to-end spoken language understanding community and us to merge the individual modules for higher system performance.
>
> **Q4: The limitations of the proposed work are not discussed. Can you describe in which cases the proposed method would fail or perform just as good as other methods?**
>
> We add the discussion about the limitations to the last section of our paper.
>
> **Q5: Is there an evaluation on time efficiency? Is the proposed method also more efficient with respect to other alternatives?**
>
> Yes. We add a section in Appendix F for the evaluation on time efficiency of all the methods. We report the average time in second (s) for the model to take one action in the iTHOR environment with the FSC dataset. We see that the slowest method is ASR+NLU+RL (ANR) which takes 0.04s for an action. This pipeline is slower than other alternatives because it contains more independent modules than other methods. The fastest method is E2E and the time efficiency for VAR++ is between those of the ANR and the E2E. VAR++ is a bit slower than the E2E because it contains an additional representation which supports self-supervised RL after the deployment. All the methods are fast enough for real-time fine-tuning and other robotic applications.
>
> **Q6: Reproducibility**
>
> We attached the code as a supplementary material. We will open-source our codebase for better reproducibility upon acceptance.
>
> **Q7: Novelty and contribution**
>
> We update our contributions in our introduction and impacts in our conclusion. We also discuss about the novelty and contribution in the "Response to all reviewers" on this webpage.

---

> > ### Comment · Reviewer_Fs4r · 2022-11-29
> > **Reply to response**
> >
> > Thank you very much for addressing my comments and for updating your manuscript. I have increased my score to 6, as I think that while the contribution may be incremental, the paper has improved since its first version, plus the results reported, together with the release of models and tasks, are interesting.

---

### Official Review · Reviewer_dfAL · 2022-10-24

**Confidence:** 3
**Clarity, Quality, Novelty And Reproducibility:** good
**Correctness:** 3
**Technical Novelty And Significance:** 3
**Empirical Novelty And Significance:** 3
**Recommendation:** 6

**Strength And Weaknesses:**

Strengths:
- Improvement over the existing VAR representation, which can be potentially helpful beyond the presented setting.
- Self supervised RL training method appears solid and effective.
- Enable robots to follow audio command is a useful and essential skill.

Weaknesses:
- Fine-tuning doesn’t seem very efficient in general: 2400 labels and 1M self-supervised RL training for fine-tuning seems a lot. Does the -user need to provide 2400 labels (image/audio pairs)? Providing the label and collecting 1M samples in the real-world seems infeasible.
- Not all images have intentions in real environments, which would lead to sparse reward problems and make the method potentially difficult to handle long-range tasks.
- No examples of handling dynamics gap were presented even though it is claimed to handle that. For robotic control tasks they are in general simple: e.g. the manipulator only needs to reach certain location without interactions like grasping. It's not clear if the method can effectively adapt to like manipulation of unseen objects.


**Summary Of The Paper:**

The paper introduced a method to learn a visual-audio conditioned control policy. The core idea is to learn a joint visual-audio representation and derive a self-supervised reward for training task policies. To acquire a high-quality visual-audio representation, they improved the existing VAR algorithm by adopting a batch-based supervised constrastive loss to improve the data efficiency. The policy reward is then defined as the difference between the embedding of the current image and the embedding of a sound signal provided at the beginning to indicate the task goal. The method was applied to two control tasks and one navigation-related task.

**Summary Of The Review:**

I think the paper is in general interesting and can be a good work for the venue, though I do have a couple concerns as listed in the weaknesses above.

In addition I also have some clarification questions for the work:
1) For E2E was it trained with 1M RL steps as well? Or it was terminated after doing a pass through the 2400 new labels?
2) For the intrinsic reward, was eq 4 or eq 5 used in the final system?
3) For the training process, how were the sound labels as well as intentions generated?

---

> ### Author Response · Authors · 2022-11-17
> **Response to Reviewer dfAL**
>
> We thank the reviewer for the comments and suggestions. We add experiments and contents to address the weaknesses and answer the  clarification questions. The modifications we made is marked by blue text in the paper.
>
> **Q1: Fine-tuning doesn’t seem very efficient in general: 2400 labels and 1M self-supervised RL training for fine-tuning seems a lot. Does the -user need to provide 2400 labels (image/audio pairs)? Providing the label and collecting 1M samples in the real-world seems infeasible.**
>
> To clarify, the users are not required to provide at least 2400 pairs or let the robot self-improve for 1M RL steps. 2400 labels and 1M self-supervised RL training were set for all the methods for a fair comparison. To support this claim, we add Table 5 showing the relationship between number of pairs and self-supervised RL steps to our paper. The results show that our method achieves decent performance with fewer than 2400 new pairs and 1M self-supervised RL steps. Please see Table 5 for more details.
>
> **Q2: Not all images have intentions in real environments, which would lead to sparse reward problems and make the method potentially difficult to handle long-range tasks.**
>
> Yes, that’s a good point. However, we would like to respectfully point out that training a VAR++ as an intrinsic RL reward is a first-of-its-kind method in command following robots. Other papers [1, 2] require laborious state measurements and extensive instrumentations if we were to fine-tune these methods. In contrast, our method provides a way to fine-tune a trained model and is intuitive to non-expert users. The effect of sparse rewards might be a remaining problem but is beyond the scope of this paper. In fact, solving sparse reward problems is also an open area of research for RL. We included this point in Sec. 5.
>
> **Q3: No examples of handling dynamics gap were presented even though it is claimed to handle that. For robotic control tasks they are in general simple: e.g. the manipulator only needs to reach certain location without interactions like grasping. It's not clear if the method can effectively adapt to like manipulation of unseen objects.**
>
> We thank the reviewer for pointing this out. We add an additional experiment in the Kuka environment to show that our method can handle dynamics gaps and adapt to unseen objects. We change the link mass, the joint friction, and PID controller’s parameters of the robot. In addition, we also replace three of the blocks to a capsule, a teddy bear, and a rabbit. Please see “Fine-tuning in new Kuka environment” in Sec. 4.4 for more details.
>
> **Q4: clarification questions for the work:**
>
> **For E2E was it trained with 1M RL steps as well? Or it was terminated after doing a pass through the 2400 new labels?**
>
> The E2E is terminated after it depletes the label quota. See “Definition of labels” in Sec. 4.4 for how to calculate the label usage. In the experiment “Fine-tuning in novel scenarios” in Sec. 4.4, all the methods are allowed to only use a fixed number of new labels. The E2E reaches the label limit much earlier than 1M RL steps because it consumes labels at every step. Our method consumes no labels during self-supervised RL and can run for as many RL steps as desired.
>
> **For the intrinsic reward, was eq 4 or eq 5 used in the final system?**
>
> We used Eq. 5 for training and Eq. 4 for fine-tuning, because we assume that the labels are sufficient during training but hard to obtain during fine-tuning, as mentioned in “Fine-tuning in novel scenarios” in Sec. 4.4.
>
> **For the training process, how were the sound labels as well as intentions generated?**
>
> We associate the intents in the sound dataset to the corresponding skills in the simulators. For example, one of the intents in FCS dataset is “activate music” which includes the sound commands like “Play, ” and “Put on the music.” We define the corresponding skill as turning on the TV in the simulator. The sound labels are indices of the tasks, but our method does not have access to sound labels.
>
> References
>
> [1] P. Chang, S. Liu, H. Chen and K. Driggs-Campbell, "Robot Sound Interpretation: Combining Sight and Sound in Learning-Based Control," in IEEE/RSJ International Conference on Intelligent Robots and Systems (IROS), 2020, pp. 5580-5587.
>
> [2] M. Shridhar, J. Thomason, D. Gordon, Y. Bisk, W. Han, R. Mottaghi, L. Zettlemoyer, and D. Fox, “Alfred: A benchmark for interpreting grounded instructions for everyday tasks,” in IEEE Computer Society Conference on Computer Vision and Pattern Recognition (CVPR), 2020.

---

> > ### Comment · Reviewer_dfAL · 2022-11-29
> > **Thanks for the response!**
> >
> > The authors' response is greatly appreciated, especially for the new experiments and code release. Thus I have increased my score to 6. However, some of my concerns still hold: the current fine-tuning, even though outperforming the baseline, is too expensive to be practically useful.

---

### Author Response · Authors · 2022-11-17
**Response to all reviewers**

We thank all the reviewers for their constructive comments. The modifications we made is marked by blue text in the paper.

**Q1: Novelty and contribution seems marginal**

We respectfully argue that the novelty of adapting (self)-supervised contrastive loss in robotics is twofold: (1) the learned representation is of higher-quality than previous triplet loss based methods, which is to our knowledge the first study in robotics and aligned with the findings in other machine learning areas such as computer vision [1] and natural language processing [2], and more importantly (2) the resulting training and fine-tuning pipelines are largely simplified from the perspective of human-robot interaction, which is one advantage of contrastive loss over triplet loss that is underexplored in other areas. More specifically, triplet loss based methods require the provision of both positive and negative pairs from robot users. In contrast, the proposed VAR++ only needs positive pairs for both training and fine-tuning stage, making the human-robot interaction more natural and efficient since a user can teach a robot as teaching a child. This is as important as the performance improvement of the control policy because such a natural human-robot interaction can promote human perception and adoption of robotic systems and marks one step towards practical social robot applications.

---

### Decision · Program_Chairs · 2023-01-20

**Decision:**

Reject

**Justification For Why Not Higher Score:**

The problem is interesting and relevant to the community, but the novelty is limited as acknowledged by the reviewers. All three reviewers read the author response and while one increased their score (to 6), they found that the authors' response improved the paper, but did not address their primary concerns.

**Justification For Why Not Lower Score:**

N/A

**Metareview: Summary, Strengths And Weaknesses:**

The paper considers the problem of robot instruction-following with visual inputs when commands are provided as audio inputs as opposed to text (i.e., typed or provided by a speech recognizer), which is typically assumed to be available by existing methods. The paper proposes the use of a contrastive loss to learn a joint audio-visual representation that is then used to define a form of intrinsic reward for policy learning. The method is evaluated in comparison to different baselines on two simulated control and one navigation task.

Instruction-following has received significant renewed interest within the robotics and the broader machine learning communities of-late. As such, the topic of the paper is of interest to many in the ICLR community. The ability to handle audio inputs as opposed to requiring that text be provided to the agent is refreshing. It opens the door to practical challenges and opportunities (e.g., handling audio observations that do not correspond to instructions) that are ignored/abstracted away by text-based methods. The novelty of this work primarily lies in replacing the loss used to train an existing audio-visual architecture with a contrastive loss. As the reviewers point out, the significance of this contribution is limited. The use of a contrastive loss to improve the quality of the joint audio-visual representation is certainly sensible, but also predictable. The paper provides a nice validation of the benefits of contrastive learning for an important problem, but without further novelty in this domain or clear implications for other domains, the contributions are limited.

**Summary Of Ac-Reviewer Meeting:**

N/A